# Integrating artificial intelligence (AI) in healthcare: advancing older adults' health management in Saudi Arabia through AI-powered chatbots



Sabah Abdullah Al-Somali

Management Information System Department, Faculty of Economics and Administration, King Abdulaziz University, Jeddah, Saudi Arabia

## ABSTRACT

**Background:** The healthcare sector is experiencing rapid digital advancements, with patients increasingly seeking quick and seamless interactions. Artificial intelligence (AI)-driven healthcare chatbots are becoming an integral part of elderly care, transforming provider-patient engagement and supporting health behavior goals tailored to individual preferences, needs, and limitations.

**Methods:** This study developed a comprehensive research framework incorporating various theoretical perspectives to explore the determinants of sustained use of AI-powered healthcare chatbots among older adults. The framework also examined the mediating influence of perceived humanness. The model was evaluated using partial least squares structural equation modeling (PLS-SEM) on cross-sectional data collected from 158 individuals aged 60 and above.

**Results:** The findings show that satisfaction with AI-powered chatbots is significantly influenced by facilitating conditions, perceived hedonic motivation, confirmation, performance expectancy, and effort expectancy. Perceived security also plays a critical role in shaping satisfaction and the intention to continue using these chatbots. Moreover, the analysis revealed that perceived humanness mediates the relationship between satisfaction and continuous use intentions among elderly users in Saudi Arabia.

**Discussion:** This research provides valuable insights into the factors influencing older adults' acceptance of AI chatbots in Saudi Arabia, particularly in the post-COVID-19 era. These findings enrich academic discourse and offer actionable recommendations for healthcare organizations adapting to the evolving digital landscape.

## INTRODUCTION

The COVID-19 pandemic abruptly compelled organizations to adopt technological innovations and applications to sustain operations and ensure business continuity. It is undeniable that the healthcare industry is experiencing rapid digital transformation and

Corresponding author
Sabah Abdullah Al-Somali,
saalsomali@kau.edu.sa

advancements in the way that healthcare services are delivered (*Sun & Zhou, 2023*). Artificial intelligence (AI), large language model (LLM), natural language processing (NLP), and conversational chatbots exemplify the numerous innovations making their way into healthcare systems, life sciences companies, providers of care and healthcare professional practices (*Davenport & Kalakota, 2019*). LLMs can process vast amounts of medical data to provide accurate responses and automate tasks like appointment scheduling and medication reminders. On the other hand, NLP facilitates chatbots in comprehending, interpreting, and responding to human language, thereby enabling more sophisticated interactions and the extraction of essential information from unstructured text (*Sindhu et al., 2024*; *Yang et al., 2024*). Generative AI chatbots, including ChatGPT, Copilot, and Gemini, are increasingly being adopted across multiple sectors. Microsoft's Copilot is thought to augment productivity by aiding users in writing, editing, and data analysis tasks. Likewise, Gemini, created by Google DeepMind, employs sophisticated algorithms for enhanced conversational abilities and knowledge acquisition. Moreover, ChatGPT has exhibited remarkable proficiency in producing human-like text for various applications, generating significant interest in technology. In fact, these Generative AI chatbots illustrate the growing adoption of AI-driven solutions across various sectors, demonstrating the adaptability and potential of generative AI in augmenting human abilities and improving user experiences.

In recent years, AI-powered chatbots and virtual assistants have become prominent, emerging in several sectors, including banking, e-commerce, healthcare, airports, insurance, and the automobile industry (*Keerthana, Vaishali & Anitha, 2024*). Chatbots deployed using AI technology (AI chatbots) are artificial intelligence conversational agents that imitate human conversation using text or voice messages. Having the ability to understand natural language and provide personalized responses, healthcare chatbots are gaining popularity due to the growing adoption of telemedicine and virtual healthcare (*Khadija, Zahra & Naceur, 2021*). The global healthcare chatbots market was valued at USD 195.85 million in 2022 and is projected to reach USD 1,168 million by 2032, with a compound annual growth rate (CAGR) of 20.1% from 2023 to 2032 (*Market.US, 2024*). Designed to provide 24/7 support to patients, these AI-powered solutions allow healthcare practitioners to concentrate on other crucial tasks and responsibilities. While AI chatbots maintain human-like interactions when simultaneously responding to a large number of users, the evidence indicates that the benefits of such technology may not be uniformly spread among all populations (*Ni et al., 2024*; *Aggarwal et al., 2023*). Specifically, it is crucial to tackle the unique challenges faced by older adults in the adoption of new technology to enable their assimilation into an information-driven society (*Shang et al., 2024*; *Győrffy et al., 2023*; *Mace, Mattos & Vranceanu, 2022*). As the global population of 60 years and older rapidly increases, it will likely exceed one billion by 2030 and reach 1.6 billion by 2050. When compared to wealthy developed states, the ageing population is currently advancing more swiftly in developing nations (*United Nation, 2023*).

In the context of the covered subject, the focus is placed on AI-powered chatbots designed to interact with senior patients, improve their healthcare experiences, and support them in achieving health goals. The perceived usefulness of chatbots depends on

several factors, such as privacy, security issues, ability to process human language, and design. As with other technological novelties, additional challenges may be associated with integrating chatbots into existing systems and processes (*Adamopoulou & Moussiades, 2020*; *Go & Sundar, 2019*). Older adults exhibit unique behavioral patterns in technology acceptance, differing significantly from other demographic groups, which often results in obstacles to fully engaging with chatbot technologies. Researchers emphasize notable disparities in motivations, cognitive abilities, attitudes, and perceptions that shape their interactions with emerging technologies (*Shang et al., 2024*; *Chen & Chan, 2011*). In a post-COVID environment characterized by the growing population of technologically versatile elderly patients, customized and tailored solutions are required to facilitate their digital engagement.

Although chatbot technology has been widely studied in healthcare settings (*Almalki & Azeez, 2020*; *Kim et al., 2021*), research focusing on AI-powered healthcare chatbots for older adults remains limited. Specifically, there is a lack of empirical investigations addressing how user satisfaction, motivation, gratification, and user experience factors shape senior users' perceptions and intentions to adopt chatbots, particularly in Saudi Arabia (*Almalki & Azeez, 2020*). This study seeks to explore the primary determinants of user satisfaction and the continued adoption of AI-powered healthcare chatbot solutions by older adults in Saudi Arabia. Digital transformation in the healthcare sector is central to the Vision 2030 framework, which underscores the role of digital health innovations and technological progress as pivotal elements in the Kingdom's development goals (*Saudi Vision, 2016*). Guided by this perspective, the research addresses the following questions:

1) What are the key factors influencing elderly users' satisfaction with, and their intention to continue using, AI-powered healthcare chatbots?
2) Does user satisfaction influence the intention to continue using AI-powered healthcare chatbots?
3) Does perceived humanness moderate the relationship between user satisfaction and the continuous intention to use AI-powered chatbots?

Building upon the provided background, the subsequent sections discuss the theoretical approach, the model framework, and the research hypotheses derived from existing literature. The Methods chapter delineates the construct measurements and data collection methodologies employed in this study. The Results chapter articulates the findings of data analysis and hypothesis testing. Finally, the discussion includes managerial implications, limitations, and potential future directions.

## AI-powered chatbots in healthcare

AI-powered chatbots provide versatile features, including real-time assistance, tailored information, and uninterrupted connectivity. Chatbots enhance patient engagement by providing appointment reminders, integrating electronic medical records (EMRs) for customized health information, and offering guidance on healthy lifestyle choices, including diet, exercise, and illness prevention. These applications collect data from diverse sources, assisting patients in managing their medical conditions with

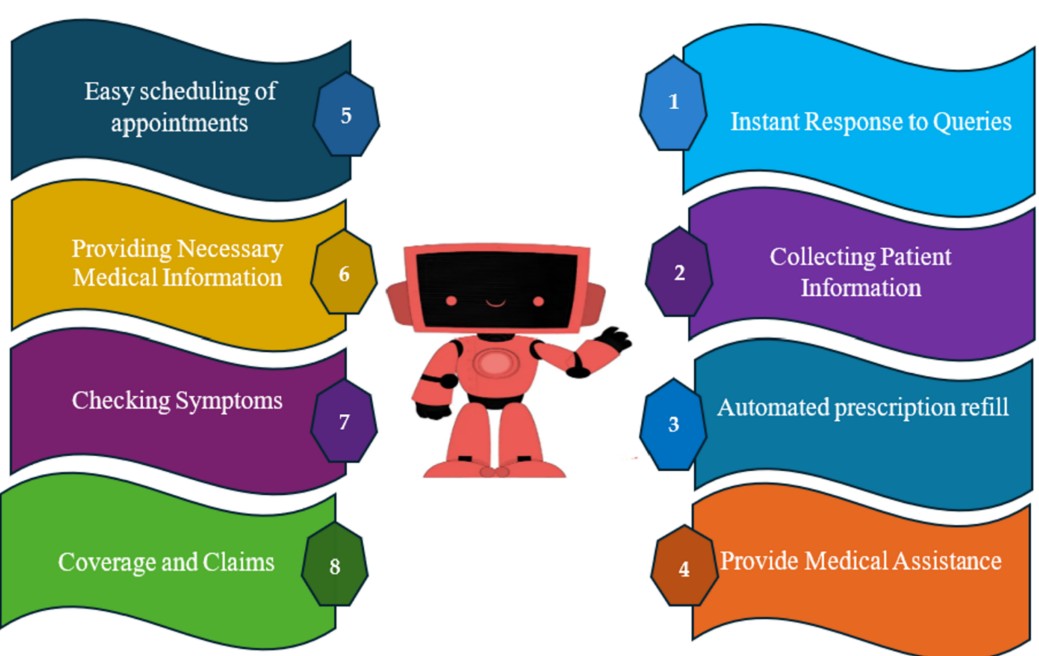

**Figure 1** Chatbots are integrated into different healthcare settings by enhancing accessibility and patient engagement.               

personalized guidance and timely reminders about medications and physical activities (*Aggarwal et al., 2023*). Moreover, these chatbots can seamlessly integrate with virtual and augmented reality platforms, delivering immersive experiences that can accelerate and enhance health behavior modifications (*Aggarwal et al., 2023*). Figure 1 illustrates examples of scenarios for using chatbots in healthcare, with diverse situations enabling the transformation of healthcare services.

The global tendency of adopting AI chatbots in healthcare reflects a growing trend across different types of healthcare systems. In the United States, AI-driven virtual health assistants have been integrated into chronic disease management programs to help with automated monitoring and behavioral coaching of patients (*Igwama et al., 2024*; *Pan et al., 2025*). In Japan, socially assistive AI chatbots and other solutions are widely considered for implementation in eldercare facilities to improve patient well-being through interactive companionship and cognitive stimulation (*Ho et al., 2023*). While these solutions address both medical and emotional needs, low-resource settings are yet to effectively embrace AI technology for primary healthcare support. Nevertheless, innovative approaches are emerging to bridge this gap. AI-driven data analytics and predictive modeling hold promise in bridging these gaps by optimizing resource allocation, enabling early disease detection, and tailoring interventions to specific geriatric populations, even in infrastructure-limited environments. (*Odionu & Ibeh, 2024*; *Zhao & Li, 2024*).

## Digital transformation and the digital divide

Digital transformation in healthcare, integrating technologies like telemedicine, AI, virtual consultations, and the Internet of Things (IoT), aims to sustain access, improve service

delivery, and enhance patient engagement (*Mulukuntla & Venkata, 2020*). However, this shift has exacerbated the digital divide among older adults, making them vulnerable due to inadequate digital literacy and limited access to reliable technologies. Defined as the disparity in access to digital technologies influenced by socioeconomic status, geographic location, and age (*Jaarsveld, 2020*), the digital divide poses significant challenges for the elderly, particularly in healthcare. Despite advancements in AI applications that enhance independence and encourage participation in care, digital exclusion remains a critical issue. Older adults face substantial barriers in using new technologies, impacting their health outcomes as more services move online (*Carrasco-Dajer et al., 2024*; *Chu et al., 2022*; *Sriwisathiyakun & Dhamanitayakul, 2022*). Addressing these barriers is crucial for fostering inclusion and enhancing the quality of life, allowing them to live independently at home longer.

A significant issue elderly people encounter when utilizing AI and chatbots is a lack of digital literacy. *Pearce (2024)* asserts that the digital literacy gap may result in dissatisfaction and disengagement, hindering older individuals from accessing essential health information or services that could enhance their well-being. Additionally, research indicates that the complexity of numerous AI-driven platforms frequently overlooks the needs of older users, leading to age-related bias and a detrimental user experience (*Chu et al., 2022*). On the other hand, trust and security apprehensions significantly contribute to the hesitance of older individuals to interact with AI and chatbots (*Iancu & Iancu, 2023*). Studies indicate that a considerable segment of this demographic not only lacks the skills required to use computers proficiently but are also reluctant to provide personal health information online due to data privacy concerns and potential security breaches (*Carrasco-Dajer et al., 2024*; *Iancu & Iancu, 2023*; *Shandilya & Fan, 2022*; *Sriwisathiyakun & Dhamanitayakul, 2022*). Addressing these concerns is essential to creating an inclusive digital health environment that enhances older adults' engagement with AI and chatbots, improving health outcomes. As healthcare providers increasingly adopt digital technologies, elderly patients need customized solutions that enhance their digital engagement and access to essential services, particularly in a post-COVID environment.

## Theoretical foundation

The progression of technology has transformed various facets of everyday life, with significant impacts on healthcare and the dissemination of information. As individuals increasingly utilize digital applications, understanding the motivations and challenges affecting technology adoption is essential. Several theoretical frameworks have emerged to elucidate this phenomenon, clarifying the intricate link between people and technology. Examining how older adults interact with digital health technologies, including AI and chatbots, represents a crucial research domain, especially as such technologies become increasingly integral to healthcare. To gain deeper insights into older adults' utilization of AI and the factors shaping their adoption of AI-driven healthcare chatbots, this study introduces a comprehensive model grounded in the expectancy confirmation model (ECM) and the unified theory of acceptance and use of technology (UTAUT2), providing a framework to analyze their engagement with these innovations.

### Unified theory of acceptance and use of technology

An advancement of the original UTAUT model (*Venkatesh et al., 2003*), UTAUT2 offers a robust framework for analyzing technology acceptance. Introduced by *Venkatesh, Thong & Xu (2012)*, UTAUT2 integrates critical components like hedonic motivation alongside the foundational constructs of performance expectancy, effort expectancy, social influence, and facilitating conditions. Designed as a versatile framework, it provides valuable insights into the evaluation of new technologies by various user groups, particularly older adults, with a focus on their behavioral intentions (*Tamilmani et al., 2021*; *Yaseen & Al Omoush, 2013*). Despite its strengths, UTAUT2 has notable limitations due to its primary focus on predicting behavioral intentions rather than directly addressing the post-adoption phase (*Tamilmani et al., 2021*). Specifically, while behavioral intention is crucial for understanding initial acceptance, understanding the sustained use and overall satisfaction among older adults presents a different challenge. The intricacies of user experience pertaining to complex technology such as chatbots require supplementary perspectives that may not be entirely captured by UTAUT2 alone. Research applying UTAUT2 in various domains, including health technology and mobile applications, increasingly acknowledged the importance of user engagement and satisfaction (*de Blanes Sebastián, Antonovica & Guede, 2023*; *Migliore et al., 2022*; *Schomakers et al., 2022*; *Tian et al., 2024*). Whereas integrating UTAUT2 with other frameworks helps address its limitations, this study utilizes the model to assess the disruptive influence of chatbots among the elderly in Saudi Arabia.

### Expectancy confirmation model

The Expectancy Confirmation Theory (ECT) is a theoretical framework that clarifies the impact of initial expectations on user satisfaction and subsequent usage behavior. Emerging from consumer behavior studies, ECM correlates consumers' confirmation with their total consuming experience and indicates the impact of these experiences on continuance intention and post-adoption perceived usefulness (*Bhattacherjee, 2001*). The significance of ECM in comprehending chatbot utilization among older users lies in its emphasis on the correlation between user expectations and pleasure. Expectations might vary markedly among populations, and for older users, considerations such as user-friendliness and perceived security are essential in their initial assessment of chatbots. *Hossain & Quaddus (2012)* emphasize that in information systems research, ECT facilitates the connection between initial acceptance and sustained engagement, rendering it particularly pertinent for comprehending the intricate behaviors of older adults in a swiftly changing technological environment. Integrating ECT into the UTAUT2 paradigm enhances our understanding of how older persons' initial expectations of chatbots affect their post-adoption experiences and plans for long-term usage. *Gupta, Yousaf & Mishra (2020)* contend that pre-adoption expectations can profoundly influence post-adoption continuation intentions. Consequently, integrating the components of UTAUT2 with ECT helps elucidate the psychological and social variables determinants influencing older individuals' views and sustained utilization of chatbot technology, yielding significant implications for the design and execution of chatbot systems specifically customized for older users.

## Conceptual model and hypotheses development

The proposed research model identifies key antecedents of chatbot utilization, primarily sourced from the ECM and UTAUT2 frameworks, with perceived humanness (HPU) serving as a mediating factor. This results in a holistic model that extends the analytical robustness and applicability of the UTAUT2 framework. It integrates several UTAUT2 variables, including social influence, perceived hedonic motivation, performance expectancy, facilitating conditions, and effort expectancy, to provide a comprehensive understanding of chatbot adoption. HPU is hypothesized to mediate the relationship between satisfaction with AI-driven healthcare chatbots and the continuous intention to use chatbots among elderly individuals in Saudi Arabia. The integration of UTAUT2 with the ECM provides a synergistic framework that improves our comprehension of technology acceptance and continuation behavior, especially regarding the utilization of chatbots by older persons. It also recognizes older users' varied and frequently intricate motivations, whose interaction with technology may be shaped by both practical needs (as delineated by UTAUT2) and emotional considerations (as outlined by ECM). By integrating variables from ECM—such as user satisfaction and confirmation—into the UTAUT2 paradigm, we can clarify how many factors interact to influence the continuous usage intentions of older individuals with chatbots. Figure 2 shows the research model.

## Perceived hedonic motivation

Perceived hedonic motivation, often described as the enjoyment derived from technology use, plays a critical role in shaping its acceptance and utilization. It captures an individual's feelings, such as enjoyment or happiness, that arise from using the technology (*Venkatesh, Thong & Xu, 2012*). Prior research has shown that when consumers see technology as enjoyable, their overall satisfaction and likelihood of continuing use significantly rise (*Al-Azawei & Alowayr, 2020*; *Meske, Junglas & Stieglitz, 2019*). *Holbrook & Hirschman (1982)* argued that hedonic motivations are essential in terms of shopping motivation and the benefits obtained from shopping. Furthermore, *Meske, Junglas & Stieglitz (2019)* conducted a study to investigate the influence of hedonic motives on the adoption of enterprise social networks, demonstrating that such motivations can foster sustained user engagement and satisfaction. Highlighting the importance of hedonic motivation and enjoyment in using chatbots, *De Cicco et al. (2021)* hypothesized that it may enhance overall user satisfaction. Building on the evaluated data, this study proposes the following research hypothesis:

**H1:** *Perceived hedonic motivation positively influences older adults' satisfaction with AI-powered healthcare chatbot applications.*

## Facilitating condition

Facilitating conditions refer to users' perception of the adequacy of technical infrastructure required to effectively support the successful use of a particular system. *Liébana-Cabanillas et al. (2018)* assert that this term embodies the belief that necessary technological and organizational infrastructures exist to enable the utilization of a specific technology

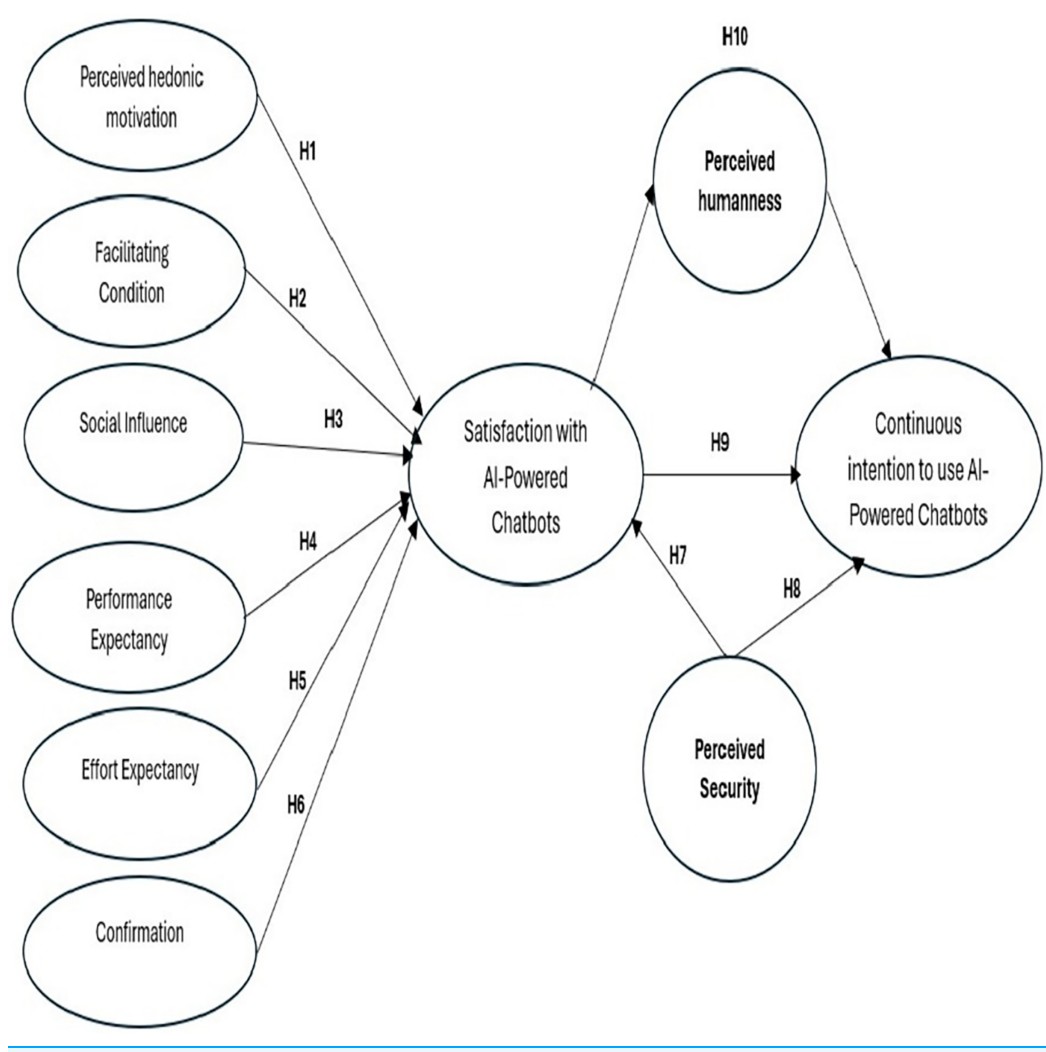

**Figure 2** The research model. 

(*Venkatesh, Thong & Xu, 2012*). Research conducted by *Amer, Almahri & Bell (2020)* indicated that favorable facilitating settings positively influence university students' intentions to utilize chatbots in the future. This highlights the significance of infrastructure preparedness and support in facilitating technology adoption. By establishing the necessary technological and organizational components, institutions can enhance user confidence and augment the probability of technology acceptance across various demographic groups. This study explicitly investigates older users' perceptions of the sufficiency of the infrastructure facilitating the usage of healthcare chatbots. Understanding whether this demographic feels the necessary resources and frameworks are available is crucial for forecasting their satisfaction with these technologies. The following research hypothesis can be proposed:

**H2**: *Facilitating conditions positively influences older adults' satisfaction with AI-powered healthcare chatbot applications.*

## Social influence

Social factors play an important role in shaping user's evaluations of technology's usefulness (*Athota et al., 2020*; *Tubaishat, 2018*). Previous research elucidates how people change their beliefs and behavior to fit into those of people around them in a bid to gain acceptance (*Tan & Liew, 2022*). Consistent with this assertion, *Shen et al. (2006)* demonstrated that social influence or perceived social pressure significantly impacts technology acceptance, meaning that when individuals observe their role models embracing a particular technology, they are likely to view this as a worthy venture due to its usefulness. *Khadija, Zahra & Naceur (2021)* observed that peer recommendations affect the perception of technology in the healthcare context, specifically, if patients see others using and endorsing chatbots, they are likely to regard these chatbots as beneficial for health management. In the realm of mobile learning, *Criollo-C et al. (2021)* assert that social influence exerts a diminished effect on the acceptance of mobile learning. On the contrary, *Bonn et al. (2016)* emphasized that social factors significantly influence individuals' perceptions and behavioral intentions about technology usage. Perceived social pressure or support can lead to heightened satisfaction as users feel more integrated into their social networks (*Pal, Funilkul & Vanijja, 2020*). For older adults, social influence and support from family, caregivers, or community members can substantially augment their confidence and satisfaction with AI-powered healthcare chatbots. The following research hypothesis can be proposed:

**H3:** *Social influence positively influences older adults' satisfaction with AI-powered healthcare chatbot applications.*

## Performance expectancy

Performance expectancy refers to an individual's belief that using a particular technology will enhance their performance or effectively meet their needs, as defined by *Venkatesh et al. (2003)*. This construct plays a pivotal role in shaping both the acceptance of a technology and the satisfaction users derive from its use. According to *Choudhury & Shamszare (2024)*, a favorable assessment of a chatbot's efficacy in executing its designated functions can result in heightened user satisfaction. Furthermore, *Abd Aziz, Aziz & Abd Rahman (2023)* underscore the notion that satisfaction can function as a significant intermediary variable that mediates the effects of performance expectancy on the comprehensive technological experience. *Rahi et al. (2019)* similarly stressed the critical nature of performance expectancy. The authors posit that when users perceive elevated performance levels, this perception correlates with enhanced satisfaction outcomes. In the context of healthcare applications, *Sitthipon et al. (2022)* discovered that users exhibit a greater propensity to engage with healthcare chatbots and applications when they believe that these technological tools will facilitate improved access to healthcare services and optimize overall health management. Based on the evaluated data, this study proposes the following research hypothesis:

**H4**: *Performance expectancy positively influences older adults' satisfaction with AI-powered healthcare chatbot applications.*

## Effort expectancy

Effort expectancy refers to the degree of simplicity and user-friendliness associated with interacting with a specific technology (*Venkatesh, Thong & Xu, 2012*). It encompasses the adaptability and user-friendliness of technology from an individual perspective. Empirical studies indicate that effort expectancy significantly affects users' intentions to embrace novel technologies. For example, *Polyportis & Pahos (2024)* underscore that effort expectancy is associated with a strong positive impact on students' ChatGPT use behavior. *Sari et al. (2024)* also concluded that not only performance but also effort expectancy are crucial determinants in the technology adoption process among Moroccan nursing students. Furthermore, *Elkhatibi, Guelzim & Benabdelouahed (2024)* researched the variables impacting the adoption of AI chatbots in the banking sector, affirming that elevated levels of effort expectancy are positively correlated with user satisfaction. The investigation carried out by *Elok & Hidayati (2021)* provides valuable insight into the close interconnection between effort expectancy and user satisfaction regarding digital wallet utilization. Their findings emphasize that effort expectancy constitutes a vital variable influencing user satisfaction. This association is particularly noteworthy within the paradigm of user engagement with digital platforms, such as AI-enhanced healthcare chatbots. Building on the evaluated data, this study proposes the following research hypothesis:

**H5:** *Effort expectancy positively influences older adults' satisfaction with AI-powered healthcare chatbot applications.*

## Confirmation

The influence of confirmation on the satisfaction derived from new technology constitutes a pivotal domain of inquiry, particularly as users increasingly interact with sophisticated solutions such as artificial intelligence and digital platforms. As articulated by *Bhattacherjee (2001)*, the affirmation of expectations has a substantial impact on sustained satisfaction and the continued utilization of technological resources. Within the realm of digital commerce, *Guo, Zhang & Xia (2023)* examined the implications of website design on customer satisfaction and loyalty. They contend that when users perceive that their experiences align with or surpass their prior expectations, there is a corresponding increase in satisfaction, which fosters enhanced loyalty. Furthermore, *Sinha & Singh (2023)* conducted a focused investigation into the engagement of elderly consumers with multichannel banking services, reevaluating the ECM to demonstrate the significance of confirmation in determining satisfaction levels. Their research suggests that for older adults, who may encounter unique usability challenges, the affirmation of expectations is a critical determinant of their overall satisfaction. Additionally, *Thong, Hong & Tam (2006)* emphasized that post-adoption beliefs, which stem from the confirmation of initial expectations, are essential for maintaining user engagement with information technology. This relationship is further corroborated by the research of *Cavallo et al. (2018)*, who investigated robotic services for older adults, discovering that perceived usefulness and satisfaction are intricately linked to the extent to which users' expectations are validated

through their experiences. Analyzing mobile instant messaging applications, *Oghuma et al. (2016)* found that the confirmation of expectations significantly influences users' intentions to persist in utilizing these services. Consequently, we postulate the following hypothesis:

**H6:** *Confirmation positively influences older adults' satisfaction with AI-powered healthcare chatbot applications.*

## Perceived security

Perceived security refers to the degree to which users believe their interactions with technology such as chatbots are secure and devoid of any inherent risk predispositions (*Shrestha et al., 2021*). Notably, the use of chatbots poses significant security threats since they frequently handle sensitive data, necessitating the implementation of rigorous security measures (*Keerthana, Vaishali & Anitha, 2024*). *Nicolescu & Tudorache (2022)* argue that threats associated with personal data protection are regarded as one of the key challenges and constraints related to the utilization of chatbots for both organizations and users. The security of healthcare chatbots is critical to protecting sensitive data, hence fostering trust among users (*May & Denecke, 2022*). When the user believes that his or her information stored on the personal and medical data is protected, the user will not have any reservations about using the chatbot as it is deemed a safe mode of communication. Additionally, implementing robust security features can eliminate users' unauthorized access to the system and limit the occurrence of interruption overall (*Athota et al., 2020*). Given the potential adverse effects of privacy and security risks, particularly in critical sectors like banking and healthcare, safeguarding user data is pivotal for achieving customer satisfaction within the emerging disruptive technology landscape (*Cheng & Jiang, 2020*; *Cox & Dale, 2001*; *Taehyee, Taekyung & Namho, 2020*). When users are assured that their interactions with the chatbot are secure, then they will spend more of their time engaging with the application and experimenting with its functionalities, hence deepening the ease-of-use construct (*Kasilingam, 2020*). Building on the evaluated data, this study proposes the following research hypotheses:

**H7:** *Perceived security positively influences older adults' satisfaction with AI-powered healthcare chatbot applications.*

**H8:** *Perceived security positively influences older adults' continuous intention to use AI-powered chatbots.*

## Satisfaction and continuous intention to use AI-powered chatbots

Satisfaction with technology markedly affects users' intentions to persist in utilizing a given product or service. Empirical research substantiates the assertion that favorable user experiences arising from satisfaction are directly associated with prolonged engagement. For example, *Alruwaili, Shaban & Elsayed Ramadan (2023)* conducted a systematic review centering on digital health interventions and their contributions to the promotion of

healthy aging. As the authors highlight that user satisfaction plays a critical role in ensuring the continued use of digital health tools, they note that when individuals recognize these technologies as beneficial and effective, they are more likely to utilize them consistently. *Cheng & Jiang (2020)* also explored the implications of AI-driven chatbots on user experience. Their results indicate that satisfaction, influenced by the perceived gratifications derived from chatbot interactions and the concomitant privacy concerns, significantly impacts users' loyalty and intent to continue employing these technologies. When users report satisfaction with their interactions, they exhibit a greater propensity to remain loyal to technology and to utilize it repeatedly. The following research hypothesis can be proposed:

**H9:** *Satisfaction with AI-powered healthcare chatbot applications positively influences older adults' continuous intention to use AI-powered chatbots.*

## The mediation effect of perceived humanness

The Humanizing Experience Theory underscores the significance of fostering a perception of humanity within digital interactions, especially *via* the use of chatbots (*Singh & Malik, 2024*). Perceived humanness is a construct commonly used in human–computer interaction (HCI), defined as the extent to which an entity is regarded as human (*Hendriks et al., 2020*). Multiple studies have reinforced the importance of human-like qualities in accepting chatbots (*Rapp, Curti & Boldi, 2021*; *Hendriks et al., 2020*; *Go & Sundar, 2019*; *Elson, Derrick & Ligon, 2018*). *Go & Sundar (2019)* contended that chatbot attributes, like friendly interface and natural language, can improve users' impressions of the chatbot as more anthropomorphic. This humanization can cultivate emotional connections, resulting in enhanced satisfaction and sustained utilization. *Lei, Shen & Ye (2021)* conducted analogous research that focused on the perceived significance of humanness in chatbot interactions. The authors revealed that users frequently associate a more anthropomorphic appearance with elevated satisfaction levels. This indicates the imperative for chatbot designers to integrate human-like characteristics to fulfill user expectations and improve the overall experience. Further highlighting the importance of chatbot humanization in customer service, *Hsu & Lin (2023)* posited that consumers' satisfaction and loyalty towards customer service chatbots are associated with perceived humanness. On the other hand, *Ding & Najaf (2024)* emphasized that chatbots with human-like characteristics might foster more engaging and relatable interactions, hence improving user satisfaction and increasing the probability of continuing usage. *Chattaraman, Kwon & Gilbert (2012)* investigated the impact of human-like interaction in virtual agents on retail websites, with a particular emphasis on elderly consumers. Their research demonstrated the significance of building virtual agents with human-like attributes to serve elder users and improve their satisfaction efficiently. As older users exhibit diverse degrees of comfort and familiarity with technology, rendering the human-like attributes of chatbots is crucial for enhancing happiness and promoting sustained usage (*Chattaraman, Kwon & Gilbert, 2012*). Through the integration of anthropomorphic attributes, including empathy and distinct personality

| Table 1 Summary for research hypotheses. |
|---|
| **Hypotheses** |
| H1 Perceived hedonic motivation positively influences older adults' satisfaction with AI-powered healthcare chatbot applications. |
| H2 Facilitating conditions positively influences older adults' satisfaction with AI-powered healthcare chatbot applications. |
| H3 Social influence positively influences older adults' satisfaction with AI-powered healthcare chatbot applications. |
| H4 Performance expectancy positively influences older adults' satisfaction with AI-powered healthcare chatbot applications. |
| H5 Effort expectancy positively influences older adults' satisfaction with AI-powered healthcare chatbot applications. |
| H6 Confirmation positively influences older adults' satisfaction with AI-powered healthcare chatbot applications. |
| H7 Perceived security positively influences older adults' satisfaction with AI-powered healthcare chatbot applications. |
| H8 Perceived security positively influences older adults' continuous intention to use AI-powered chatbots. |
| H9 Satisfaction with AI-powered healthcare chatbot applications positively influences older adults' continuous intention to use AI-powered chatbots |
| H10 Perceived humanness serves as a significant mediator in the relationship between user satisfaction and the continuous intention to utilize these technologies among elderly users. |

traits, chatbots can potentially enhance user interaction engagement. Building on the evaluated data, this study proposes the following research hypothesis:

**H10 :** *Perceived humanness serves as a significant mediator in the relationship between user satisfaction and the continuous intention to utilize these technologies among elderly users.*

Table 1 provides summary for research hypotheses.

## MATERIALS AND METHODS

### Research design

Healthcare chatbot systems offer personalized and effective health recommendations, with accuracy that closely resembles human medical expertise. Accordingly, this research seeks to identify the elements that encourage older patients to consistently engage with AI-driven healthcare chatbot technologies. The conceptual framework integrates perspectives from motivation theory, UTAUT, and ECT to provide a comprehensive understanding of these factors. To achieve the research objectives, a quantitative approach was adopted, focusing on the continuous usage of AI-powered healthcare chatbots. Data were gathered through a structured survey questionnaire, a method well-suited for systematically capturing behaviors, attitudes, perceptions, and beliefs, thereby enabling the extrapolation of findings.

### Data collection procedures

An exploratory, cross-sectional survey design was employed to examine the adoption of AI-driven healthcare chatbot solutions among elderly individuals in Saudi Arabia. This research was carried out under the auspices of the Faculty of Economics and Administration at King Abdulaziz University. The Research Ethics Committee (REC) reviewed and approved the research (reference number: REC 1/5). Participants provided informed consent before participation to confirm their voluntary involvement and awareness of data confidentiality. Data collection was conducted through an anonymous

survey distributed *via* Google Forms; an online platform known for its efficiency in managing electronic surveys. Moreover, Google Forms has a user-friendly design and robust data management capabilities. In alignment with our framework, participant eligibility was restricted to those aged 60 and above who utilized healthcare chatbots solutions. A convenience sample was obtained using a snowball sampling technique facilitated through WhatsApp, the most commonly used social networking tool in Saudi Arabia, during June and July 2024. The researchers utilized their professional and personal networks to disseminate the survey invitation. All responses were anonymized, and no personally identifiable data were collected.

This sampling method was chosen for two primary reasons. First, WhatsApp's widespread use in Saudi Arabia makes it an ideal platform for reaching the target demographic. Second, the snowball approach enabled the study to reach older participants by leveraging familial and social connections. Interested respondents were guided to the survey platform, where they could review the study's objectives.

Correspondingly, respondents were required to provide their consent and confirm their age prior to enrolling in the survey. The survey invitation explicitly stated that the participation is voluntary and participation in the survey will imply providing informed consent to guarantee the confidentiality of participants. Participants indicated their consent by selecting "Agree" to proceed to the survey questions.

A total of 162 responses were collected, with a response rate of 74%. After removing four submissions due to missing information or uniform responses across all questionnaire items, the final sample size comprised 158 participants, representing 71% of the initial pool. Participant anonymity was assured, and no answers were deemed right or wrong. The survey design prioritized clarity and precision, with ambiguous terminology eliminated to ensure comprehensibility and accuracy in responses.

## Data analysis techniques

Partial least squares structural equation modeling (PLS-SEM) was employed for data analysis, employing the Smart PLS 4.0 software suite (*Silva, Shojaei & Barbosa, 2023*; *Cheah et al., 2018*). This method, which applies partial least squares estimation, is particularly suitable for handling complex models comprising multiple constructs and indicators (*Guenther et al., 2023*). A key advantage of PLS-SEM is its ability to analyze data effectively without requiring a large sample size due to its component-based approach (*Lohmöller, 2013*).

## Survey instruments

To ensure methodological rigor and relevance to the study's context, the measurement scales for each construct were adapted from established scholarly sources. For instance, 'Perceived Security' and 'Perceived Hedonic Motivation' were assessed using three items each, derived from *Noor, Rao Hill & Troshani (2022)* and a combination of *Iancu & Iancu (2023)* and *O'Brien (2010)*, respectively. Four items measuring 'Facilitating Condition' were informed by the works of *Venkatesh et al. (2003)*, *Venkatesh, Thong & Xu (2012)*.

'Social Influence' relied on two items adapted from *Venkatesh et al. (2003)*, *Venkatesh, Thong & Xu (2012)*, while 'Performance Expectancy' was assessed with three items sourced from *Ni et al. (2024)* and *Venkatesh et al. (2003)*, *Venkatesh, Thong & Xu (2012)*. 'Effort Expectancy' included two items based on *Slade, Williams & Dwivedi (2013)* and *Venkatesh, Thong & Xu (2012)*. Similarly, 'Confirmation' utilized three items from *Aggarwal et al. (2023)* and *Lee, Hsieh & Hsu (2011)*, and 'Satisfaction' was measured with four items following *Bhattacherjee (2001)*. 'Perceived Humanness' included three items adapted from *Go & Sundar (2019)*, while 'Continuous Intention & Immersion to use AI-Powered Chatbots' employed three items primarily based on *Al-Sharafi et al. (2023)* and *Thong, Hong & Tam (2006)*. All constructs were measured using a 5-point Likert scale ranging from 1 (strongly disagree) to 5 (strongly agree). Further details on the measurement instruments and their sources can be found in Appendix A.

# RESULTS

## Demographic information

The demographic data presented in Table 2 reveals that 46.2% ($n = 73$) of respondents were male, while 53.8% ($n = 85$) were female. Notably, 67.7% ($n = 107$) of participants reported having bachelor's degrees, indicating a predominantly highly educated cohort. The statistical analysis of income distribution indicates that a notable proportion of participants, specifically 46.8%, receive annual earnings that are less than 16,000 SAR. Simultaneously, 29% of the individuals surveyed are categorized within the moderate-income bracket, which ranges from 16,000 to 26,000 SAR, whereas 15.8% of respondents indicate that their earnings surpass 26,000 SAR. Furthermore, 2.5% of the respondents opt not to disclose their income information, and 5.7% express uncertainty regarding their income levels. Regarding their reasons for using healthcare chatbots, 31% ($n = 49$) utilized the technology for medical assistance, 30.4% ($n = 48$) for prescription refills, and 5.1% ($n = 8$) for scheduling appointments. Finally, 54.4% ($n = 86$) of the participants indicated that they accessed healthcare chatbot from their smartphones, and 45.6 % ($n = 72$) used their personal computers to access healthcare chatbot.

## Measurement model assessment

This research employed PLS-SEM as a data analysis tool to evaluate convergent validity, discriminant validity, and reliability (*Hair et al., 2019*). PLS-SEM, a variance-based approach, is particularly suited for estimating structural models and optimizing explained variance in dependent variables (*Aburumman et al., 2022*). The evaluation included internal consistency reliability through Cronbach's alpha (CA) and composite reliability (CR). As shown in Table 3, all CA values exceeded 0.60, and CR values surpassed 0.70, indicating robust internal consistency (*Hair et al., 2019*). A Cronbach's alpha value of 0.60 or higher was regarded as acceptable for each construct's internal consistency. Convergent validity was assessed *via* average variance extracted (AVE), with all constructs achieving AVE values greater than 0.50, confirming the scales' validity and reliability.

**Table 2 Demographic information.**

| Variables | Categories | Frequency | Percent | Valid percent | Cumulative percent |
|---|---|---|---|---|---|
| Gender | Male | 73 | 46.2 | 46.2 | 46.2 |
| | Female | 85 | 53.8 | 53.8 | 100 |
| Education | High school | 4 | 2.5 | 2.5 | 2.5 |
| | Bachelor's degree | 107 | 67.7 | 67.7 | 70.3 |
| | Master | 45 | 28.5 | 28.5 | 98.7 |
| | Doctorate | 2 | 1.3 | 1.3 | 100.0 |
| Income | Prefer not to say | 4 | 2.5 | 2.5 | 2.5 |
| | Less than 16,000 SAR | 74 | 46.8 | 46.8 | 49.4 |
| | Between 16,000 SAR and 26,000 SAR | 46 | 29.1 | 29.1 | 78.5 |
| | More than 26,000 SAR | 25 | 15.8 | 15.8 | 94.3 |
| | Do Not Know | 9 | 5.7 | 5.7 | 100.0 |
| Reasons for using healthcare chatbot | To schedule appointments | 8 | 5.1 | 5.1 | 5.1 |
| | To get prescription refill | 48 | 30.4 | 30.4 | 35.4 |
| | To get medical assistant | 49 | 31.0 | 31.0 | 66.5 |
| | To get information about disease prevention | 28 | 17.7 | 17.7 | 84.2 |
| | To inquiry about insurance claims | 25 | 15.8 | 15.8 | 100.0 |
| Healthcare chabot access | From PC | 72 | 45.6 | 45.6 | 45.6 |
| | From smart phone/tablet | 86 | 54.4 | 54.4 | 100.0 |

**Table 3 Convergent validity and reliability.**

| Constructs | Items | Factor loadings | Cronbach's alpha (CA) | Composite reliability (CR) | Average variance extracted (AVE) |
|---|---|---|---|---|---|
| Perceived hedonic motivation (HED) | HED1 | 0.805 | 0.780 | 0.808 | 0.690 |
| | HED2 | 0.828 | | | |
| | HED3 | 0.858 | | | |
| Facilitating condition (FAC) | FAC1 | 0.915 | 0.937 | 0.938 | 0.841 |
| | FAC2 | 0.920 | | | |
| | FAC3 | 0.925 | | | |
| | FAC4 | 0.907 | | | |
| Social influence (SOC) | SOC1 | 0.915 | 0.867 | 0.948 | 0.879 |
| | SOC3 | 0.960 | | | |
| Performance expectancy (PER) | PER1 | 0.860 | 0.868 | 0.887 | 0.79 |
| | PER2 | 0.938 | | | |
| | PER3 | 0.866 | | | |
| Effort expectancy (EFF) | EFF1 | 0.718 | 0.605 | 0.779 | 0.694 |
| | EFF2 | 0.935 | | | |
| Confirmation (CON) | CON1 | 0.940 | 0.887 | 0.889 | 0.817 |
| | CON2 | 0.942 | | | |
| | CON3 | 0.947 | | | |
| | CON4 | 0.934 | | | |

| Table 3 (continued) | | | | | |
|---|---|---|---|---|---|
| Constructs | Items | Factor loadings | Cronbach's alpha (CA) | Composite reliability (CR) | Average variance extracted (AVE) |
| Perceived security (SEC) | SEC1 | 0.947 | 0.928 | 0.930 | 0.874 |
| | SEC2 | 0.925 | | | |
| | SEC3 | 0.933 | | | |
| Satisfaction (SAT) | SAT1 | 0.906 | 0.887 | 0.889 | 0.817 |
| | SAT2 | 0.938 | | | |
| | SAT3 | 0.866 | | | |
| Continuous intention to use AI-powered chatbots (COT) | COT1 | 0.935 | 0.885 | 0.891 | 0.815 |
| | COT2 | 0.940 | | | |
| | COT3 | 0.829 | | | |
| Perceived humanness (HPU) | HPU1 | 0.935 | 0.950 | 0.951 | 0.909 |
| | HPU2 | 0.964 | | | |
| | HPU3 | 0.961 | | | |

## Discriminant validity

To ensure constructs were distinct, the study assessed discriminant validity using the Fornell–Larcker criterion and Heterotrait-Monotrait Ratio (HTMT). Table 4 shows that HTMT values were below 0.85, confirming adequate differentiation between constructs (*Aburumman et al., 2022*). Furthermore, Table 5 demonstrates that the square root of each construct's AVE exceeded its correlations with other constructs, affirming discriminant validity.

## Model fitness

Model fitness, reflecting the alignment of the conceptual model with observable data, was assessed using SRMR. The SRMR value of 0.077, below the threshold of 0.08, confirms satisfactory model fitness (*Hair et al., 2021*). As detailed in Table 6, $R^2$ values indicate that independent variables explained 76.5% of variance in satisfaction with AI-powered chatbots, while satisfaction, perceived security, and perceived humanness accounted for 62.2% of the variance in continuous intention to use chatbots. Perceived humanness alone contributed to 39.7% of variance in satisfaction. These findings validate the model's predictive power and accuracy (*Miles, 2005*).

## Structural model assessment

This study employed the bootstrapping technique, generating 5,000 sub-samples to evaluate the research hypotheses and assess the statistical significance of path coefficients (*Hair et al., 2019*). The structural model was then analyzed to test the hypotheses and calculate effect sizes (*Hair, Howard & Nitzl, 2020*). This assessment focused on determining the model's predictive accuracy and the relationships among constructs. A 5% significance level with a 95% confidence interval was applied, requiring *p*-values below 0.05

**Table 4 Discriminant validity: Heterotrait–Monotrait ratio (HTMT) matrix.**

| Construct | CON | COT | EFF | FAC | HED | HPU | PER | SAT | SEC | SOC |
|---|---|---|---|---|---|---|---|---|---|---|
| CON | | | | | | | | | | |
| COT | 0.677 | | | | | | | | | |
| EFF | 0.655 | 0.716 | | | | | | | | |
| FAC | 0.692 | 0.789 | 0.795 | | | | | | | |
| HED | 0.506 | 0.571 | 0.647 | 0.525 | | | | | | |
| HPU | 0.652 | 0.682 | 0.477 | 0.661 | 0.430 | | | | | |
| PER | 0.638 | 0.757 | 0.804 | 0.771 | 0.502 | 0.611 | | | | |
| SAT | 0.740 | 0.786 | 0.843 | 0.842 | 0.519 | 0.685 | 0.835 | | | |
| SEC | 0.603 | 0.779 | 0.719 | 0.738 | 0.710 | 0.520 | 0.689 | 0.815 | | |
| SOC | 0.519 | 0.579 | 0.319 | 0.513 | 0.390 | 0.459 | 0.331 | 0.522 | 0.514 | |

**Note:**
CON, confirmation; COT, Continuous intention & immersion to use AI-powered chatbots; HPU, perceived humanness; EFF, effort expectancy; FAC, facilitating condition; HED, perceived hedonic motivation; PER, performance expectancy; SAT, satisfaction; SOC, social influence; SEC, perceived security.

**Table 5 Discriminant validity: Fornell–Larcker.**

| Construct | CON | COT | EFF | FAC | HED | HPU | PER | SAT | SEC | SOC |
|---|---|---|---|---|---|---|---|---|---|---|
| CON | **0.941** | | | | | | | | | |
| COT | 0.621 | **0.903** | | | | | | | | |
| EFF | 0.531 | 0.550 | **0.833** | | | | | | | |
| FAC | 0.655 | 0.720 | 0.620 | **0.917** | | | | | | |
| HED | 0.440 | 0.474 | 0.472 | 0.464 | **0.831** | | | | | |
| HPU | 0.622 | 0.626 | 0.401 | 0.625 | 0.388 | **0.954** | | | | |
| PER | 0.588 | 0.674 | 0.615 | 0.703 | 0.433 | 0.569 | **0.889** | | | |
| SAT | 0.681 | 0.698 | 0.661 | 0.769 | 0.444 | 0.630 | 0.749 | **0.904** | | |
| SEC | 0.570 | 0.712 | 0.571 | 0.690 | 0.612 | 0.492 | 0.625 | 0.740 | **0.935** | |
| SOC | 0.474 | 0.513 | 0.249 | 0.469 | 0.324 | 0.417 | 0.306 | 0.473 | 0.468 | **0.938** |

**Notes:**
CON, confirmation; COT, continuous intention & Immersion to use AI-powered chatbots; HPU, perceived humanness; EFF, effort expectancy; FAC, facilitating condition; HED, perceived hedonic motivation; PER, performance expectancy; SAT, satisfaction; SOC, social Influence; SEC, perceived security.
The bold numbers in the diagonal are the square root of AVE of each construct, and other numbers are correlations between constructs.

and $t$-values above +1.96. Table 7 provides the results of hypothesis testing, and Fig. 3 presents the structural equation model.

H1 confirms a significant positive relationship between perceived hedonic motivation and satisfaction with AI-powered chatbots ($\beta = 0.110$, $t = 1.972$, $p = 0.049$). Similarly, H2 indicates that facilitating conditions significantly enhance satisfaction ($\beta = 0.190$, $t = 2.077$, $p = 0.038$). In contrast, H3 shows that social influence does not significantly impact satisfaction ($\beta = 0.089$, $t = 1.742$, $p = 0.081$). Performance expectancy exerts a strong positive effect on satisfaction (H4: $\beta = 0.262$, $t = 3.570$, $p = 0.000$). Effort expectancy and confirmation also demonstrate positive influences (H5: $\beta = 0.164$, $t = 2.625$, $p = 0.009$;

**Table 6 R-square and model fit results.**

| Variable | R-squared (R²) | Standardized root mean square residual (SRMR)-estimated model |
| --- | --- | --- |
| Satisfaction with AI-Powered chatbots (SAT) | 0.765 | 0.076 |
| Perceived humanness (HPU) | 0.397 | |
| Continuous intention to use AI-powered chatbots (COT) | 0.622 | |

**Table 7 Direct and mediating effects.**

| Hypotheses | Direct/indirect path | Beta value (β) | t-value | P value | VIF | Confidence interval (95%) bias corrected | Hypothesis validation |
| --- | --- | --- | --- | --- | --- | --- | --- |
| H1 | HED → SAT | 0.110 | 1.966 | 0.049 | 1.676 | [0.002 to −0.222] | Supported |
| H2 | FAC → SAT | 0.190 | 2.077 | 0.038 | 2.996 | [−0.001 to 0.012] | Supported |
| H3 | SOC → SAT | 0.089 | 1.749 | 0.080 | 1.476 | [−0.001 to −0.01] | Not supported |
| H4 | PER → SAT | 0.262 | 3.565 | 0.000 | 2.391 | [−0.002 to −0.120] | Supported |
| H5 | EFF → SAT | 0.164 | 2.624 | 0.009 | 1.982 | [0.001–0.041] | Supported |
| H6 | CON → SAT | 0.158 | 1.991 | 0.046 | 2.079 | [0.004 to −0.001] | Supported |
| H7 | SEC → SAT | 0.287 | 2.985 | 0.003 | 2.681 | [−0.003 to 0.102] | Supported |
| H8 | SEC → COT | 0.414 | 4.612 | 0.000 | 2.216 | [−0.01 to 0.394] | Supported |
| H9 | SAT → COT | 0.207 | 2.003 | 0.045 | 2.790 | [0.009–0.214] | Supported |
| H10 | SAT → HPU → COT | 0.184 | 3.117 | 0.002 | 1.662 | [0.004–0.065] | Supported |

H6: $\beta = 0.158$, $t = 1.996$, $p = 0.045$). Perceived security significantly impacts both satisfaction (H7: $\beta = 0.287$, $t = 2.985$, $p = 0.003$) and continuous intention to use AI-powered chatbots (H8: $\beta = 0.414$, $t = 4.612$, $p = 0.000$). Satisfaction positively influences continuous intention (H9: $\beta = 0.207$, $t = 2.003$, $p = 0.045$). Finally, H10 reveals that perceived humanness mediates the relationship between satisfaction and continuous intention ($\beta = 0.184$, $t = 3.117$, $p = 0.002$). Collinearity diagnostics confirmed that all variance inflation factor (VIF) values were below 3.3, indicating no multicollinearity issues (*Hair et al., 2021*; *Miles, 2005*).

## DISCUSSION

The pandemic served as a catalyst compelling organizations to adopt new technological innovations and solutions to survive and continue operations. The healthcare industry is undergoing rapid digital transformation and advancements in the way that healthcare services are delivered (*Park & Kim, 2023*). This research aimed to explore the critical factors driving satisfaction and the intention to continue using AI-powered healthcare chatbot solutions among older adults in Saudi Arabia. With the rapid proliferation of technology, older adults face substantial challenges in adoption, making the study of their technology acceptance a pressing priority. While the impact of age on technology acceptance has been widely investigated in commercial sectors, limited research has addressed this issue within healthcare settings (*Slade, Williams & Dwivedi, 2013*). A

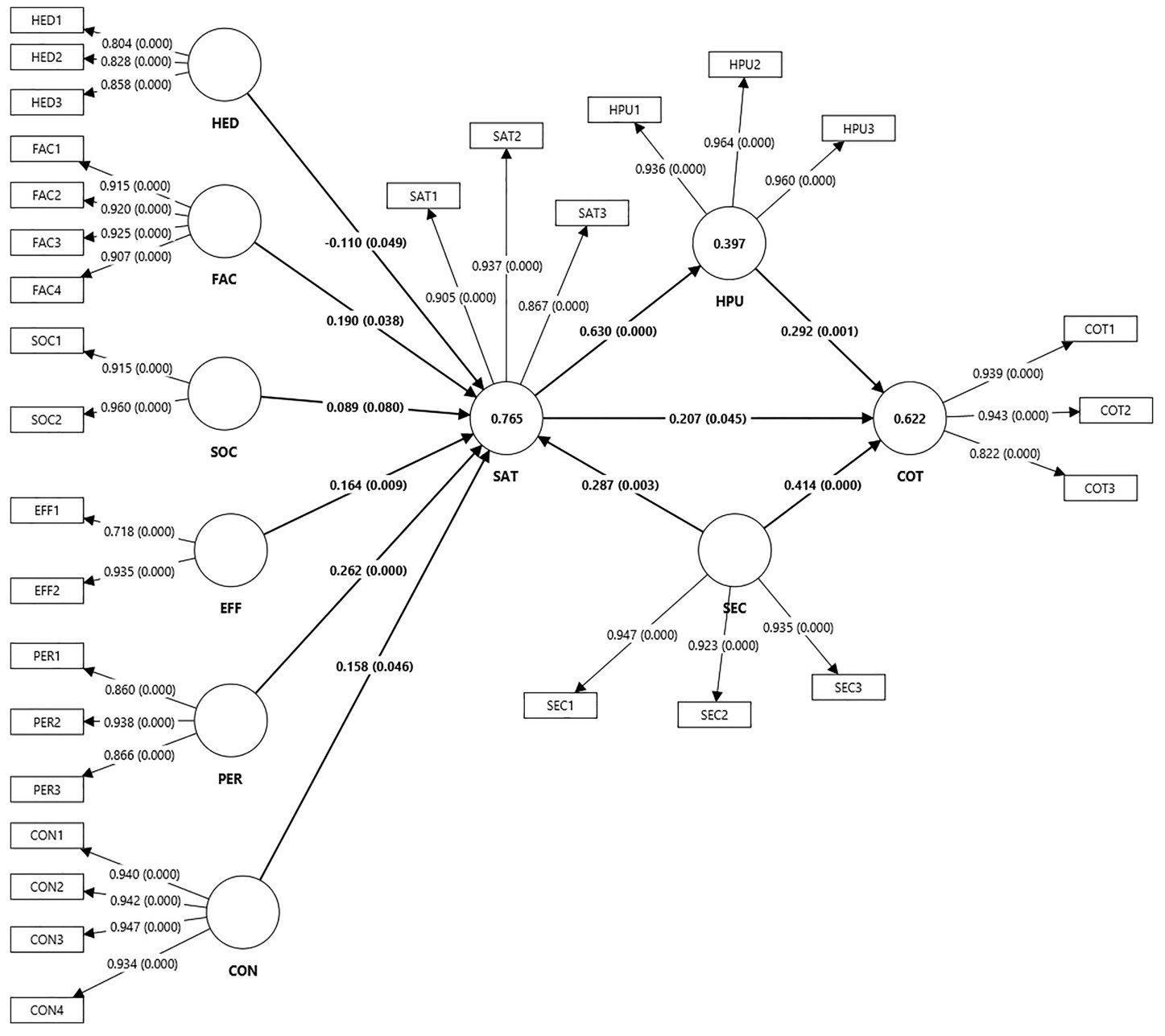

**Figure 3 The structural equation model.**

comprehensive understanding of technology adoption among older adults is vital for fostering their inclusion in the information-driven society and improving their quality of life through technological innovations.

The study findings indicate that perceived hedonic motivation positively and significantly affects satisfaction with AI-powered chatbots. These results are consistent with prior research emphasizing that enjoyment and perceived hedonic value are pivotal in the adoption and sustained use of emerging technologies (*Akdim, Casaló & Flavián, 2022*). Recent research has also shown that hedonic motivation, which entails both pleasure and

enjoyment tied to technology use, is a major determinant of the acceptance rates and satisfaction levels among different user demographics, including the elderly (*Al-Azawei & Alowayr, 2020*). In their research, *Shang et al. (2024)* indicated that older adults perceived greater hedonic motivation and enjoyment, satisfaction, and acceptance with text feedback than without Intelligent Virtual Assistants. Overall, higher levels of hedonic motivation typically led to a positive attitude toward using chatbot technology, resulting in sustained engagement and positive user experiences. This underscores the need for chatbots designed to address the emotional and psychological preferences of older adults, ensuring both functionality and enjoyment in their use.

Conversely, the results indicate that perceived security substantially enhances older adults' satisfaction and their intention to continue using AI-powered chatbots. These findings align with prior research by *Athota et al. (2020)*, *Cheng & Jiang (2020)*, *Cox & Dale (2001)*, and *Taehyee, Taekyung & Namho (2020)*, highlighting the pivotal role of perceived security in sustaining engagement with emerging technologies like AI-driven chatbots. When older adults perceive that their data is secure, they may be more inclined to rely on chatbots for ongoing health management, regularly seeking assistance for their healthcare needs.

While social influence is often considered a key factor in technology adoption across various demographic groups, it appears to be less significant for older adults. The analysis reveals that social influence exhibits a positive path coefficient of 0.089 with a $p$-value exceeding 0.05, leading to the rejection of the hypothesis that social influence has a meaningful impact on satisfaction. This corresponds with prior studies, which have also highlighted that social influence does not impact consumer satisfaction (*Ashfaq et al., 2020*). A potential reason for this insignificant effect of social influence is that older users often emphasize the usefulness and functionality of technology rather than external considerations like peer recommendations or social connections (*Iancu & Iancu, 2023*). Considering that healthcare is a very personal area, older individuals may prioritize the effectiveness and reliability of AI chatbots in addressing their health requirements over the opinions of friends or family. Furthermore, older people frequently encounter reduced social engagement with their peer groups, leading to less dependence on social influence.

This divergence from traditional technology adoption models, such as UTAUT2, aligns with findings indicating that social influence plays a comparatively diminished role among older adults engaging with digital health technologies (*Busch et al., 2021*; *Townsend, Chen & Wuthrich, 2021*). Instead, intrinsic motivation and perceived usefulness often outweigh external social factors, particularly when digital tools serve a direct health-related purpose (*Moore et al., 2021*). Indeed, acknowledging that social influence has limited impact on older adults' satisfaction with AI healthcare chatbots enables a shift in focus toward improving the user experience and functional effectiveness of these technologies. By focusing on usability and relevance, developers can enhance older individuals' satisfaction and empowerment in healthcare management.

The findings of this study emphasize that user satisfaction is a pivotal factor for the effective implementation and ongoing usage of AI chatbots within the studied demographic. Prior studies have shown that older adults are more inclined to adopt and

find satisfaction in new technologies when their unique needs and preferences are adequately addressed (*Alruwaili, Shaban & Elsayed Ramadan, 2023*). This finding highlights the necessity of developing AI chatbots that prioritize user experience, particularly for vulnerable populations like older adults, ultimately leading to continuous intention to utilize healthcare chatbots. Moreover, we demonstrated that facilitating conditions positively affect older adult satisfaction with AI-powered chatbots. Among older users, confidence in the adequacy of organizational and technical support plays a decisive role in their willingness to adopt and engage with innovative technologies. An environment that offers sufficient support—such as easy access to devices, training on how to use chatbots, and ongoing technical assistance—promotes greater confidence among older adults in interacting more confidently with these AI systems. When older users are confident in their ability to obtain assistance, their general pleasure with technology improves, reducing anxiety and fostering a more favorable user experience (*Venkatesh, Thong & Xu, 2012*).

The research identified perceived humanness as a key mediating factor linking user satisfaction with the continued intention to utilize AI-powered chatbots among elderly individuals. By integrating human-like traits into chatbot design, such as empathetic language, a congenial tone, and personalized responses, it becomes possible to bridge the gap between users and technology, rendering conversations more relatable and engaging (*Elson, Derrick & Ligon, 2018*; *Go & Sundar, 2019*; *Hendriks et al., 2020*). To improve the perceived humanness of healthcare chatbots, developers should prioritize natural language processing capabilities that facilitate more fluid and conversational interactions. Moreover, integrating accessibility features, such as voice interaction and clear language, would enhance the technology's usability for older users. By considering the views of older adults, healthcare organizations can develop chatbots that address clinical requirements while promoting a more human-centered approach to digital health solutions.

The findings, though directly relevant to Saudi Arabia's digital health transformation, should be assessed in comparison to global shifts in elderly AI adoption. In technologically advanced economies, AI healthcare chatbots are already seamlessly integrated into telemedicine frameworks and assistive eldercare. For example, both Japan and Germany leverage pre-existing high digital literacy and robust infrastructure (*Aggarwal et al., 2023*; *Shang et al., 2024*). By contrast, infrastructural limitations and digital literacy gaps in developing regions necessitate alternative engagement models, such as voice-assisted AI or mobile-first chatbot solutions. The latter are often tailored to local linguistic and technological ecosystems (*Carrasco-Dajer et al., 2024*).

Cultural expectations further shape adoption. In predominantly individualistic Western countries, the entrenchment of digital autonomy creates favorable conditions for widespread adoption of self-service AI tools. However, physician-mediated digital interventions are still preferred in collectivist cultures, such as the Middle East, China, and Latin America (*Mace, Mattos & Vranceanu, 2022*). Perceived humanness also exerts varied influence, with regions characterized by strong digital trust embracing AI-driven empathy modeling. Conversely, skepticism persists in societies with privacy concerns and skepticism towards foreign technologies (*Go & Sundar, 2019*; *Győrffy et al., 2023*). In this context, there is a

demand for localized solutions accounting for cultural expectations, privacy laws, and technological infrastructure. These insights frame the study's results as they underscore the need for adaptive, context-aware AI deployment that would ensure inclusive digital healthcare for aging populations worldwide.

## Theoretical implication

The present work significantly contributes to scientific literature in the healthcare and technology context. This research offers three notable contributions to the theoretical understanding and modeling of technology acceptance. First, the findings provide valuable enhancements to the theoretical foundation concerning new technology adoption. By leveraging insights from motivational theory, UTAUT, and ECT, this study examines the key factors influencing the sustained use of intelligent healthcare chatbots by older adults. Second, the study identified several factors including perceived security, perceived humanness, performance expectancy, and effort expectancy that significantly enhanced existing literature, indicating that more human-like interactions can foster a stronger emotional connection, resulting in increased satisfaction and continued use of AI-powered healthcare chatbots among older adults. Finally, the results underscore the importance of regional context, particularly in Saudi Arabia. The study offers empirical evidence from both demographic and geographic perspectives that have hitherto been neglected. By situating the findings within Saudi Arabia's unique cultural and socio-economic context, this study lays the groundwork for future investigations into the adoption of emerging technologies in similar settings. This regional emphasis strengthens the theoretical foundation of technology acceptance. It offers valuable avenues for future research that can inform practical applications in healthcare technology and act as a comparative benchmark for further research in other emerging markets.

## Practical implications

The study results inform actionable recommendations for organizations and policymakers aiming to utilize AI-powered healthcare chatbots to improve older adults' engagement and satisfaction. First, key predictors of satisfaction and continued use, identified in the study, underscore the importance of designing chatbot interfaces that prioritize user accessibility, interactivity, and clarity. Healthcare providers should focus on creating chatbots that seamlessly integrate into older adults' existing digital habits and preferences, promoting both ease of use and enjoyable experience. Second, the study highlights that social influence does not significantly affect the intention to continue using AI-powered chatbots. This suggests that marketing strategies should shift away from emphasizing social aspects and instead focus on individual benefits and direct user experiences. Highlighting the functional benefits of chatbots, particularly their capacity to deliver precise and timely health information, can prove more impactful in encouraging user adoption. Third, healthcare organizations should prioritize the development of AI-powered chatbot with intuitive, user-friendly, interfaces. Simplifying navigation and minimizing the steps required to access information can significantly enhance user experience, making the

technology more approachable for older adults. Similarly, ensuring that chatbots are easily integrated into existing healthcare systems and workflows can reduce the perceived complexity for both healthcare providers and patients. Seamless integration with electronic health records and other digital health tools can facilitate easier access to information and improve overall sustainability.

## Research limitations and future directions

Despite its significant theoretical and managerial contributions, this study acknowledges several limitations. Namely, the study's sample size of 158 healthcare chatbot users aged over 60 from Saudi Arabia, while informative, might not reflect the full range of diversity within the broader elderly population. Future investigations should consider utilizing larger sample sizes to improve the applicability of the findings across other countries in the region. Additionally, the reliance on a convenience sample distributed *via* WhatsApp introduces potential selection bias, as older adults who are more digitally literate or comfortable with mobile technology may be overrepresented. Since this sampling approach may exclude individuals with lower digital proficiency, it limits the generalizability of the findings to all elderly populations. From another standpoint, the dependence on self-reported data could introduce biases, such as recall inaccuracies or the inclination to provide socially desirable responses, which may affect data reliability. With these biases being potentially further compounded by the self-selection effect inherent in digital survey distribution, careful interpretation of results is warranted. To address these limitations, future studies should explore alternative recruitment methods by including offline surveys and hybrid data collection strategies.

Furthermore, cross-sectional nature of this research restricts the ability to draw conclusions about causative relationships between the studied factors and the ongoing use of healthcare chatbots. To address this, longitudinal research designs are recommended for future studies to better capture the dynamic relationships over time. Experimental approaches could further enhance understanding by isolating causal effects of chatbot adoption through controlled interventions. One of the primary recommendations is to measure behavioral changes in response to specific chatbot features or personalized AI-driven interactions.

As such, future scholars must consider incorporating mixed-methods approaches, combining qualitative and quantitative data, to address these biases and offer a more well-rounded perspective on user behavior. The study's scope is also limited to certain variables, potentially overlooking other significant factors that could influence chatbot use. Future research could examine additional influencing factors, including but not limited to privacy concerns, trust, and health literacy. Studies exploring AI-driven chatbot interventions in clinical settings or comparing their effectiveness across different healthcare models could provide deeper insights into their real-world applicability and patient outcomes. Investigating the prolonged effects of chatbot usage and the practical difficulties associated with integrating AI technologies into healthcare systems could also provide deeper insight into the associated benefits and challenges.

## CONCLUSIONS

In conclusion, this study has demonstrated that older adults exhibit unique behavioral patterns in technology acceptance, influenced by their cognitive abilities, motivations, attitudes, and perceptions regarding new technologies. In particular, older adults' acceptance of AI-powered healthcare chatbots is significantly shaped by effort expectancy, perceived facilitating conditions, hedonic motivation, performance expectancy, and perceived security. This underscores the distinct challenges and opportunities in adopting technology across different demographic cohorts. Additionally, the mediating role of perceived humanness in enhancing continuous intention to use these technologies indicates the importance of designing AI interfaces that are intuitively human-like. These insights deepen our grasp of the intricate dynamics between senior populations and emerging technologies, indicating that the adoption of AI in healthcare settings can potentially improve the well-being and digital engagement of older adults. Scholars are encouraged to delve into the sustained effects of these technologies on older demographics as well as to assess how various variables might impact the latter's acceptance levels. The continuous evolution of technology, especially in the aftermath of the COVID-19 pandemic, poses both obstacles and prospects for increasing the interaction of elderly users with healthcare technologies, emphasizing the importance of adapting AI tools to the needs of this age group as a critical area of ongoing research.

## ACKNOWLEDGEMENTS

I extend my heartfelt gratitude to all the participants in this study for their invaluable insights and willingness to share their experiences, which greatly enriched this research. Their time and effort were instrumental in bringing this study to fruition. I also sincerely appreciate the editorial team at PeerJ Computer Science for their unwavering support and guidance throughout the submission process, ensuring a smooth review and publication journey.

### Funding

This research work was funded by Institutional Fund Project under grant no. (GPIP: 1476-245-2024). The study was technically and financially supported by the Ministry of Education and King Abdulaziz University, DSR, Jeddah, Saudi Arabia. The funders had no role in study design, data collection and analysis, decision to publish, or preparation of the manuscript.

### Grant Disclosures

The following grant information was disclosed by the authors:
Institutional Fund Project: GPIP: 1476-245-2024.
Ministry of Education and King Abdulaziz University, DSR, Jeddah, Saudi Arabia.

## Competing Interests

The author declares that they have no competing interests.

## Author Contributions

- Sabah Abdullah Al-Somali conceived and designed the experiments, performed the experiments, analyzed the data, performed the computation work, prepared figures and/or tables, authored or reviewed drafts of the article, and approved the final draft.

## Ethics

The following information was supplied relating to ethical approvals (*i.e.*, approving body and any reference numbers):

The research ethics committee in the Faculty of Economics and Administration at king Abdulaziz university granted the Ethical approval to carry out the study in Saudi Arabia (Ethical Application Reference Number: REC 1/5).

## Data Availability

The raw measurements are available in the Supplemental File.

## Supplemental Information

Supplemental information for this article can be found online at http://dx.doi.org/10.7717/peerj-cs.2773#supplemental-information.

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
