# Peer review of "Integrating artificial intelligence (AI) in healthcare: advancing older adults’ health management in Saudi Arabia through AI-powered chatbots"

_PeerJ Computer Science, doi:10.7717/peerj-cs.2773_

## Round 0.1 · original submission · Minor Revisions

Regarding the reviewers’ comments, please note that Reviewer 1’s request for additional bibliographic citations is not mandatory and these citations have been deleted. You may assess the relevance of the rest of the suggestions and decide whether to incorporate them at your discretion. If you choose not to include them, no further justification is required.
Reviewer 2 has suggested some minor revisions, which are summarized below:
• Provide more details on informed consent, data protection and the Research Ethics Committee (REC) approval.
• Include an explicit discussion on potential biases introduced by the snowball sampling approach via WhatsApp.
• Expand the discussion on the broader applicability of the findings beyond Saudi Arabia and their relation to global trends in elderly digital health adoption.
• Improve the readability of Figure 5 to enhance clarity.

Reviewer 1 ·

Basic reporting

Reference Report
Title: Integrating AI in Healthcare: Advancing Older Adults' Health Management through AI-Powered Chatbots
Submitted to: PeerJ Computer Science
Manuscript ID: #109852
Comments:
This work focuses on developing a framework for an AI-powered chatbot designed for seniors in Saudi Arabia. After reviewing this manuscript, I have the following concerns:
1. Title Accuracy: The current title does not accurately reflect the scope of the work. It should include keywords such as "framework" and "Saudi Arabia."
2. Introduction Section: The authors should expand the discussion on the application of large language models (LLMs) and natural language processing (NLP) in chatbots. Recent references should be included.
3. Popular AI Chatbots: Besides ChatGPT, other popular generative AI chatbots like Copilot and Gemini should be mentioned.
4. Workflow Diagram: The authors should include a workflow diagram of the framework to illustrate the construction of the chatbot for seniors. Additionally, they should explain why the chatbot is specifically designed for seniors and discuss the potential effects and impacts if used by non-senior adults.
5. Hypotheses Summary: It would be beneficial to include a table summarizing all the hypotheses (H1 – H10) in this study.
6. Ethical Considerations: Ethical issues are crucial for AI-powered chatbots. The authors should address this topic, referencing significant works.
7. Future Work: Since this chatbot is currently designed for seniors in Saudi Arabia, the authors should discuss potential future work to expand the study globally.

Experimental design

Please refer to 1.

Validity of the findings

Please refer to 1.

Additional comments

Nil

Reviewer 2 ·

Basic reporting

Clarity and Language: The paper is well-written in professional and clear English, with minor areas that could benefit from slight grammatical refinement for better readability.
Introduction & Background: The introduction provides a strong contextual background for the research, outlining the role of AI chatbots in elderly healthcare and positioning the study within the literature.
Literature Referencing: The study is well-referenced, incorporating relevant and recent literature. However, some citations could be expanded upon to include more diverse global perspectives beyond Saudi Arabia.
Figures and Tables: Figures and tables are used effectively to present findings. However, clarity in figure captions and descriptions can be improved to make them more self-explanatory.
Raw Data Availability: The manuscript mentions raw data collection and analysis, but it should explicitly state whether datasets are available for verification.

Experimental design

Originality & Research Scope: The study contributes to the growing field of AI in healthcare by examining factors affecting older adults’ adoption of AI-powered chatbots, particularly in Saudi Arabia.
Research Question & Knowledge Gap: The research question is well-defined and aligned with an identified gap in understanding the continued use of AI chatbots among elderly users.
Methodology:
• The study applies Partial Least Squares Structural Equation Modeling (PLS-SEM) on a sample of 158 older adults (60+ years).
• The survey-based approach and hypothesis-driven model are appropriate for exploring technology adoption factors.
• A limitation is the sample size (158 participants), which, while sufficient for PLS-SEM, may limit generalizability beyond the studied population.
Replication Feasibility: The methodology is detailed enough for replication, but including the survey instrument in an appendix or supplementary material would enhance transparency.

Validity of the findings

Statistical Analysis:
• The measurement model assessment (Cronbach’s alpha, composite reliability, AVE) is robust.
• Discriminant validity is tested using the Fornell-Larcker criterion and HTMT, confirming construct validity.
• The structural model analysis is appropriately conducted using bootstrapping with 5,000 resamples.
Findings & Interpretation:
• The study finds that perceived security, performance expectancy, and hedonic motivation strongly impact chatbot satisfaction among older adults.
• Perceived humanness mediates the link between satisfaction and continued use.
• Social influence was found not significant, which contradicts some previous studies in technology adoption, warranting further discussion.
Conclusion Alignment: The conclusions align well with the findings and emphasize theoretical contributions and practical implications.

Additional comments

1. Expand the Discussion on Cross-Cultural Relevance
The findings are highly relevant to Saudi Arabia’s healthcare transformation, but how might they apply to other regions?
A comparison with global trends in elderly digital health adoption would add value.
2. Enhance Clarity in Figures & Tables
While the figures are useful, some labels and descriptions could be clearer and more detailed.
Consider revising Figure 1 and Table 5 to improve readability.
3. Address Sample Representation & Generalizability
The study is based on a convenience sample distributed via WhatsApp.
This could introduce selection bias, as tech-savvy older adults may be overrepresented.
Discussing potential biases more explicitly would strengthen the credibility of the findings.
4. Provide More Details on Ethical Considerations
The manuscript notes Research Ethics Committee (REC) approval, but more details on informed consent and data protection should be provided.
5. Clarify the Limitations Section Further
While limitations are mentioned, elaborating on future research directions (e.g., longitudinal studies, experimental approaches) would be valuable.

---

## Round 0.2 · accepted · Accept

The author has thoroughly addressed all the reviewers' comments in their revised manuscript. I have reviewed the updated version and found that the revisions have been made to a satisfactory standard. The manuscript now meets the necessary requirements, and I consider it ready for publication.